# Validation of EuroSCORE II, ACEF Score, CHA_2_DS_2_-VASc, and CHA_2_DS_2_-VA in Patients Undergoing Left Main Coronary Artery Angioplasty: Analysis from All-Comers BIA-LM Registry

**DOI:** 10.3390/jcm13226907

**Published:** 2024-11-16

**Authors:** Emil Julian Dąbrowski, Paweł Kralisz, Konrad Nowak, Kamil Gugała, Przemysław Prokopczuk, Grzegorz Mężyński, Michał Święczkowski, Sławomir Dobrzycki, Marcin Kożuch

**Affiliations:** Department of Invasive Cardiology, Medical University of Białystok, 15-089 Białystok, Poland

**Keywords:** coronary artery disease, percutaneous coronary interventions, revascularization, intravascular ultrasound, optical coherence tomography

## Abstract

**Background**: Simple surgical and clinical risk scores are useful in mortality prediction. **Aims**: The study’s aim was to validate three scores in real-world registry of percutaneous coronary intervention (PCI) for the left main coronary artery (LMCA). **Methods**: All data were obtained from the BIA-LM Registry. Discrimination and calibration of EuroSCORE II, ACEF, CHA_2_DS_2_-VASc, and CHA_2_DS_2_-VA were assessed with receiver operating characteristic (ROC) curves analysis and Hosmer–Lemeshow (HL) test. **Results**: The final cohort included 851 patients, median age was 71, and 156 patients had history of previous coronary artery bypass grafting (CABG). Median EuroSCORE II, ACEF, CHA_2_DS_2_-VASc, and CHA_2_DS_2_-VA were 3.1% (IQR 5.4%), 1.56 (IQR 0.9), 4 (IQR 2), and 4 (IQR 2), respectively. In the short- (30 days) and long-term (mean 4.1 years), there were 27 and 318 deaths. In short-term, EuroSCORE II showed the best discrimination in the overall population and subgroup with unprotected LMCA [area under the curve (AUC) 0.804, 95% CI 0.717–0.890 and AUC 0.826, 95% CI 0.737–0.913, respectively, *p* < 0.001 for comparisons with other models), with the best cut-off value at 7.1%. In long-term observation, EuroSCORE II and ACEF showed good predictive value (overall population: AUC 0.716, 95% CI 0.680–0.750 and AUC 0.725, 95% CI 0.690–760, respectively). In short- and long-term observation, EuroSCORE II and ACEF showed poor calibration (HL test *p* < 0.05) as compared to CHA_2_DS_2_-VASc (HL test *p* = 0.40 and 0.18). **Conclusions**: EuroSCORE II showed good mortality prediction in short-term observation; however, its predicted risk should be interpreted with caution due to poor calibration. ACEF and EuroSCORE II may be useful in long-term mortality prediction.

## 1. Introduction

Ischemic heart disease (IHD), most commonly caused by coronary artery disease (CAD), has been the leading cause of death worldwide for over 20 years, especially in developing countries. Presumably, due to progress in both pharmacological treatment and coronary revascularization techniques, the rate of deaths due to IHD has declined in high-income countries [1].

For years, coronary artery bypass grafting (CABG) was the mainstay of coronary revascularization. However, since the early 2000s, with the progress in percutaneous coronary intervention (PCI) device technology, intravascular imaging, and procedural techniques, the indications for percutaneous treatment have expanded. Recent years’ research questioned the rationale behind coronary artery revascularization in chronic coronary syndromes (CCS), suggesting no survival benefit in such patients [2,3]. Studies investigating invasive or conservative treatment mostly did not include patients with left main coronary artery (LMCA) stenosis—in such cases, when not revascularized, 3-year mortality reaches up to 37% which is related with even 100% of jeopardized myocardium [4].

Since 2009, LMCA disease treatment options include CABG or PCI, depending on the extent of CAD reflected by the SYNTAX score [5], left ventricular ejection fraction (LVEF), and comorbidities [6]. In order to facilitate the decision, many risk stratification tools have been developed, proposing the inclusion of numerous variables, such as sex, race, primary payor, complete blood count parameters, patient’s mobility, pulmonary disease, urgency of operation, or peripheral artery disease (PAD). Among the clinical prediction models (CPM), some stand out above the rest due to their simplicity, including the European System for Cardiac Operative Risk Evaluation (EuroSCORE) II, CHA_2_DS_2_-VASc, and age, creatinine, and ejection fraction (ACEF) [7,8,9].

EuroSCORE II was developed in 2012 for peri-operative risk stratification in patients undergoing cardiac surgery as an update of EuroSCORE I and logistic EuroSCORE calculators [9]. Its recalibration led to the better performance, while remaining inclusive and user-friendly. CHA_2_DS_2_-VASc score, although developed primarily and widely adopted for stroke prediction in patients with nonvalvular atrial fibrillation (AF), was validated several times in the setting of various groups of patients [8]. Recently, 2024 European Society of Cardiology guidelines advocated for the removal of the sex category in the calculator, introducing CHA_2_DS_2_-VA score [10]. Lastly, developed in 2009, adhering to the principle of Ockham’s razor, ACEF score was principally used for risk of death prediction in patients undergoing elective CABG, where it proved better performance than most of the complex CPMs [7].

However, whether these risk scores may have short- and long-term prognostic value in patients undergoing PCI for significant LMCA stenosis remains unknown. Therefore, the aim of this study was to assess the relation between EuroSCORE II, CHA_2_DS_2_-VASc, CHA_2_DS_2_-VA, ACEF, and mortality in the overall population of patients undergoing LMCA angioplasty and stratified on the severity and extent of CAD in a large all-comers single-centre registry from Poland.

## 2. Materials and Methods

### 2.1. Design

The present study analyzes data collected retrospectively from the BIA-LM Registry, whose design was extensively provided before [11]. In brief, the registry is a single-centre database of LMCA PCI performed in the Department of Invasive Cardiology, Medical University of Bialystok, Poland from 27 December 2008 to 21 February 2022. After the exclusion of duplicates, patients referred for surgical or conservative treatment and, with missing ≥1 of key score predictors, a final cohort of 851 individuals undergoing LMCA angioplasty were included in the analysis (Figure 1). All analyzed scores were recalculated for every patient enrolled in the study using the EuroSCORE II interactive calculator (available at https://www.euroscore.org, accessed on 27 August 2024); CHA_2_DS_2_-VASc as the sum of 1 point for diagnosis of heart failure, hypertension, age between 65 and 74 years, diabetes mellitus, vascular disease, and female gender; and 2 points for age ≥ 75 years and prior stroke or transient ischemic attack. CHA_2_DS_2_-VA was used as the sum of 1 point for diagnosis of heart failure, hypertension, age between 65 and 74 years, diabetes mellitus, and vascular disease; and 2 points for age ≥ 75 years and prior stroke or transient ischemic attack. ACEF was utilized as age divided by ejection fraction plus one if serum creatinine was ≥2 mg/dL [7,8,9].

Analyses were performed for the overall population and patients with unprotected LMCA (i.e., with no previous CABG) and protected LMCA (patients with a history of previous CABG). Further calculations were performed based on the severity of CAD. Patients were divided into four subgroups: isolated LMCA narrowing, with additional 1-, 2- or 3-vessel disease (1-VD, 2-VD, 3-VD, respectively).

### 2.2. Endpoints and Definitions

The analyzed endpoint was survival at the longest available follow-up. Data on mortality was obtained for all of the patients from the Centre for Information Technology, Minister of Digital Affairs, Poland and are valid as for 13 June 2022.

The significance of atherosclerotic lesions was based on coronary angiography, according to the pertinent definitions—diameter stenosis ≥ 50% in LMCA and 70% in other major arteries visualized in coronary angiography or fractional flow reserve ≤0.8 or diagnosis based on intravascular imaging techniques [12].

### 2.3. Statistical Analyses

Distributions of all variables were assessed with the Shapiro–Wilk test. Baseline variables were presented as a number (N) of occurrences (%). Categorical variables were compared using chi-square test or Fisher exact test in cases when the expected count of variable was <5. Continuous variables were summarized as mean and standard deviation (SD) if normally distributed; non-normal distributions were summarized as median and interquartile range (IQR) and compared with Student’s *t* test or Wilcoxon rank-sum test as appropriate.

Predictive value of EuroSCORE II, CHA_2_DS_2_-VASc, and ACEF score on all-cause mortality was assessed with receiver operating characteristic (ROC) curves analysis by computing the area under the curve (AUC). Calibration of models was appraised by Hosmer–Lemeshow test and calibration plot methodology (predicted probability of the observed (O) vs. expected (E) proportion).

For all analyses, the level of statistical significance was set at *p* < 0.05. All analyses were performed with Stata/SE 18.0 for Mac (StataCorp, College Station, TX, USA).

### 2.4. Ethical Considerations

The study was approved by the Bioethics Committee of the Medical University of Białystok, Białystok, Poland (approval no. APK.002.78.2022 obtained on 10 February 2022) and adheres to Helsinki Declaration as revised in 2013.

## 3. Results

### 3.1. Baseline Characteristics

The final cohort included 851 patients undergoing PCI for LMCA stenosis, including 156 (18%) after previous CABG (Figure 1). The median age was 71 (IQR 16) and 73% were male. The median EuroSCORE II, ACEF, CHA_2_DS_2_-VASc, and CHA_2_DS_2_-VA scores in the overall population were 3.1% (IQR 5.4%), 1.56 (IQR 0.9), 4 (IQR 2), and 4 (IQR 2), respectively. In subgroup analysis, patients with protected as compared to unprotected LMCA disease had significantly higher EuroSCORE II values [5.3% (IQR 7.7%) vs. 2.6% (IQR 4.2%), *p*-value < 0.001] with no differences in other analyzed models. Detailed baseline characteristics and median scores are presented in Table 1.

### 3.2. Discrimination

In the short-term follow-up (30 days) we observed 27 deaths (3.2%), while in the long-term (total time at risk 1,266,932 days, average 49.6 months) 318 (37.4%) of the analyzed patients died.

In the overall cohort, EuroSCORE II as compared to ACEF and CHA_2_DS_2_-VASc provided the best discriminative performance for the 30-day mortality (*p* = 0.03 and *p* < 0.001). In the long-term, EuroSCORE II and ACEF both were good mortality predictors (*p* = 0.53 for comparison) with better performance than CHA_2_DS_2_-VASc (*p* < 0.001 for both two model comparisons).

In patients with unprotected LMCA, EuroSCORE II showed better discriminative performance than ACEF and CHA_2_DS_2_-VASc in short-term (*p* = 0.051 and *p* < 0.001), while in the long-term, EuroSCORE II and ACEF were better than CHA_2_DS_2_-VASc (*p* < 0.001 for both two model comparisons). In the subgroup of post-CABG patients, EuroSCORE II and ACEF provided moderate 30-day mortality discriminative performance, as opposed to poor CHA_2_DS_2_-VASc discrimination. In the long-term, all of the analyzed CPMs provided acceptable discriminative performance. There were no significant differences between CHA_2_DS_2_-VASc and CHA_2_DS_2_-VA in the short-term (*p* = 0.24), while in the long-term, CHA_2_DS_2_-VA showed better performance (*p* = 0.02). Detailed information on the analyses are presented in Figure 2 and Figure 3 and Table 2.

In the overall population and subgroup of patients with unprotected LM, Youden’s index analysis showed the best cut-off value for 30-day mortality prediction of ACEF at 1.99 and 1.65, EuroSCORE II at 7.1% in both subgroups, and CHA_2_DS_2_-VASc at 7 points in both subgroups. In the long-term follow-up, optimal cut points in overall population and unprotected LMCA population were defined as 1.68 and 1.57 for ACEF, 3.2% and 2.4% for EuroSCORE II, and 5 points for CHA_2_DS_2_-VASc in both subgroups.

In the subanalysis based on CAD severity, in 30-day observation, EuroSCORE II showed the best discrimination in all of the subgroups, outperforming other calculators especially in patients with isolated LMCA lesion and concomitant 3-vessel disease. On the other hand, in the long-term follow-up, ACEF provided the best discriminative performance in all of the subgroups, with significant differences between AUCs in patients with concomitant 1- and 2-vessel disease. Details on comparisons are presented in Appendix A.

### 3.3. Calibration

In the overall population and subgroup of patients with unprotected LM, the Hosmer–Lemeshow tests analysis showed CHA_2_DS_2_-VASc and CHA_2_DS_2_-VA 30-day calibration, as opposite to EuroSCORE II and ACEF. In the long-term, both scores also showed the best calibration among the analyzed calculators. ACEF and EuroSCORE II overestimated mortality in the lower-risk groups, especially in the short-term and tended to underestimate mortality in higher-risk groups in the long-term observation. Detailed information regarding the models’ calibration is provided in Table 3 and Appendix A and Figure 4.

In subgroup analysis based on angiographical characteristics, all of the analyzed CPMs showed good calibration in 30-day observation. In long-term follow-up, ACEF was well-calibrated among all of the subgroups. EuroSCORE II failed Hosmer–Lemeshow test (*p* = 0.047) in the group of patients with isolated LMCA lesion and CHA_2_DS_2_-VASc showed poor calibration in the subgroup with concomitant 3-vessel disease (Hosmer–Lemeshow test *p* = 0.03). Details are presented in Supplementary Appendix A.

## 4. Discussion

To the best of our knowledge, the current study is the largest investigation of the short- and long-term prognostic impact of CPMs in patients with LMCA disease treated percutaneously. The main findings and clinical implications of this all-comers registry analysis are the following: (1) EuroSCORE II provides good predictive value in short- and long-term observation in patients with unprotected LMCA; (2) good long-term performance of the ACEF score suggests that universal three factors may act as a surrogate for comorbidity burden and general health status; (3) EuroSCORE II and ACEF showed poor calibration, suggesting they may not be reliable for identifying high-risk individuals; (4) the performance of CPMs differ when stratified on angiographical severity of CAD; (5) this is the first study to compare CHA_2_DS_2_-VASc with CHA_2_DS_2_-VA in patients with CAD and the updated score showed better long-term mortality prediction; (6) CHA_2_DS_2_-VASc showed good calibration, suggesting that it may indicate high-risk patients on a populational but not on an individual level.

A number of CPM s were IId in order to facilitate decision-making in patients with CAD undergoing PCI, including Mayo Clinic and NCDR, or combining anatomic conditions with clinical data SYNTAX II [13,14,15]. However, the inclusion of many variables leading to higher accuracy at the expense of complexity and less based on usefulness, limits the clinical utility. Our study included arguably three of the most commonly used cardiovascular clinical prediction models—EuroSCORE II, CHA_2_DS_2_-VASc, and ACEF. Although the models primarily were developed for mortality prediction in patients undergoing cardiac surgery and for risk of stroke assessment, they were previously validated in various clinical settings outside of their initial purposes, showing mixed results and applicability. However, knowing the gap in evidence for their clinical usefulness in the subgroup of patients with LMCAD, we addressed this issue in the current analysis.

EuroSCORE II was developed in 2012 as an update to the well acknowledged EuroSCORE. Its recalibration was based on 22,381 patients from 43 countries, including 16 non-European. Since the publication, it was externally validated over 65 times in various subsets of patients, including those with LMCAD [16]. Zhao et al. investigated the predictive value of EuroSCORE II in a cohort of patients with 3-vessel disease and/or LMCAD [17]. In their analysis, although EuroSCORE II provided good predictive value (AUC 0.761, 95% CI 0.711–0.813), it was not superior to logistic EuroSCORE or ACEF. Staudacher and colleagues analyzed predictors of 2-year survival in 142 patients with acute coronary syndrome undergoing LMCA PCI. Authors found that an AUC of 0.83 EuroSCORE II was a better predictor than SYNTAX and SYNTAX II scores (AUC 0.63 and 0.68, respectively), concluding that clinical data models may be more relevant than anatomical complexity scores when guiding the optimal method of treatment [18]. In the review of external validations of cardiovascular CPMs, the median validation AUC was 0.76 (IQR 0.68–0.81) [16]. Of note, EuroSCORE II was the third most validated CPM, following logistic and additive EuroSCORE.

Our results support the aforementioned studies, suggesting good EuroSCORE II discriminative power, showing that the performance is consistent after stratification on the history of previous CABG and CAD severity in short- and long-term follow-up. When it comes to calibration, mortality overestimation in the lower-risk groups may be clinically relevant. It is plausible that some of the patients who deferred from CABG may not be revascularized percutaneously due to the perception of high-mortality risk—a situation which may have important consequences, knowing the unfavourable prognosis amongst the patients with significant LMCA stenosis.

Developed by Lip et al., CHA_2_DS_2_-VASc is a simple model for the assessment of stroke and thromboembolic events in patients with nonvalvular AF [8]. Recent studies investigated the potential usefulness of the score in predicting adverse cardiovascular outcomes in various settings, including patients with coronary artery disease undergoing percutaneous revascularization. Wang et al. investigated the 1-year predictive value of CHA_2_DS_2_-VASc score in 2533 patients treated with PCI [19]. The registry included 3.9% of patients with a history of AF, 3.4% with LMCA stenosis, and only 33% had CHA_2_DS_2_-VASc score ≥ 3 points. In the model’s validation, AUC for all-cause mortality was 0.610, which was higher than for cardiac death, myocardial infarction, repeat revascularization or stroke (AUC 0.569, 0.539, 0.518 and 0.586, respectively). Authors associated the higher predictive value for all-cause mortality with the previous reports suggesting that higher CHA_2_DS_2_-VASc may anticipate stent thrombosis and no-reflow phenomenon [20,21]. Another Chinese study including 3295 patients, of which 4.5% had AF history and 46% had ≥3 points in CHA_2_DS_2_-VASc, reported similar findings (AUC 0.65, 95% CI 0.62–0.69) [22]. Interestingly, in contrast to both aforementioned studies, BIA-LM registry consisted of 6-fold higher rate of patients with AF and as many as 87% with CHA_2_DS_2_-VASc score ≥3, yet the population included a significantly higher rate of patients referred for elective procedure. The calculator did not show superiority in any of the analyzed subgroups when investigating mortality prediction, and AUCs were similar to those reported in studies. The performance of CHA_2_DS_2_-VA was similar to CHA_2_DS_2_-VASc; however, we observed better discrimination of the updated score in long-term observation. Both models showed good short- and long-term calibration in most of the subgroups, suggesting that thromboembolic risk scores actually indicate co-morbidity burden severity, which may be useful when assessing low- or high-adverse event risk patients, especially on the populational level.

Adhering to the principle of Ockham’s razor, including only three variables, the ACEF score provided better performance than most of the complex CPMs in elective CABG [7]. The median validation AUC of 0.74 calculated from 26 studies was equal to the development value [16]. Its performance in patients undergoing PCI was assessed for the first time in the population of LEADERS trial [23]. In this analysis, with a c-statistic value of 0.727 for 1-year predicting cardiac death, ACEF outperformed SYNTAX score and modified clinical SYNTAX score (AUC 0.647 and 0.710, respectively). The predictive accuracy of the score in patients undergoing bifurcation PCI was investigated in the I-BIGIS registry population [24]. The model’s performance in such patients was satisfactory in both 30-day and 24-month mortality (AUC 0.82, 95% CI 0.77–0.87 and 0.77, 95% CI 0.74–0.81, respectively). Capodanno et al. investigated the performance of angiographical, clinical, and combined models on the population of CUSTOMIZE registry undergoing LMCA PCI and CABG [25]. In the analysis of 2-year primary composite endpoint (defined as cumulative of sudden death, fatal myocardial infarction, or death secondary to heart failure), ACEF provided the best prognostic accuracy for patients undergoing surgery (AUC 0.741, 95% CI 0.650–0.832), while in those undergoing PCI, ACEF and EuroSCORE yielded the worst performance (AUC 0.687, 95% CI 0.591–0.783 and 0.688, 95% CI 0.592–0.785, respectively). On the other hand, stand-alone angiographical and combined scores were proved to be the most useful in risk stratification in angioplasty; therefore, authors opted for the use of SYNTAX as the preferable decision-making tool in LMCAD. However, an all-cause mortality analysis of this registry revealed that ACEF and EuroSCORE II provide moderate-to-good discrimination in patients undergoing percutaneous treatment (AUC 0.718, 95% CI 0.644–0.792 and AUC 0.691, 95% CI 0.613–0.769, respectively). Moreover, ACEF was the best calibrated CPM in the PCI arm (Hosmer–Lemeshow *p*-value 0.56). Our results are in line with those findings, suggesting that in patients undergoing LMCA angioplasty, SYNTAX and combined scores may be helpful in the prognostication of various middle-term cardiovascular outcomes, but in the prediction of presumably the most important clinical outcome, i.e., early-, middle-, and long-term all-cause mortality, the simple bedside-friendly ACEF or EuroSCORE II may provide the best results.

The phenomenon of good ACEF and EuroSCORE II performance in patients with LMCAD may be associated with a few explanations. First of all, due to the risk of overfitting and collinearity, the inclusion of more variables does not always translate to better model’s performance. Ranucci et al. tested the efficacy of the early EuroSCORE and its derivative models with a limited number of variables in 11,150 patients undergoing cardiac surgery [26]. The exclusion of 12 variables resulted in a five-factor model that had better calibration and clinical performance than EuroSCORE. This may be especially relevant when assessing risk in post-CABG patients—previous cardiac surgery is a factor strongly impacting higher scores. Moreover, it suggests that the three factors act as a proxy for general health status and the co-morbidity burden of patients. Secondly, EXCEL and NOBLE trials provided contradictory results in terms of revascularization strategy superiority when considering composite endpoints, but they were consistent in the analysis of all-cause mortality rates as there were no differences in the 30 days and the 3-years follow-ups. It suggests that scores, developed primarily for the cardiac surgery patients, may be the suitable for the highest-risk PCI patients [27].

## 5. Limitations

Despite the relatively large group of patients included, our study has several limitations that are needed to be addressed, some of which were described earlier [11]. In order to obtain the long-term follow-up data, the current all-comers registry covers a timeframe that resulted in the inclusion of 7% of patients that received bare-metal stents which are no longer in use. Whether it might have had an impact on our findings is unknown. Our study analyzed all-cause mortality; therefore, no conclusions regarding the prognostic impact of CPMs on major cardiac and cerebrovascular events could be drawn. The analysis is based on a single-centre registry that is run in a large tertiary hospital; therefore, potential confounding factors may include a higher risk of patients referred for a procedure, which may be reflected by a relatively high rate of rotational atherectomy use.

## 6. Conclusions

EuroSCORE II yielded the best performance for short-term mortality prediction in the overall population and amongst most of the analyzed subgroups while the ACEF score was a stronger long-term mortality predictor. Both scores showed poor calibration, especially overestimating mortality in lower risk subgroups, which may cause deferral from LMCA PCI.

## Figures and Tables

**Figure 1 jcm-13-06907-f001:**
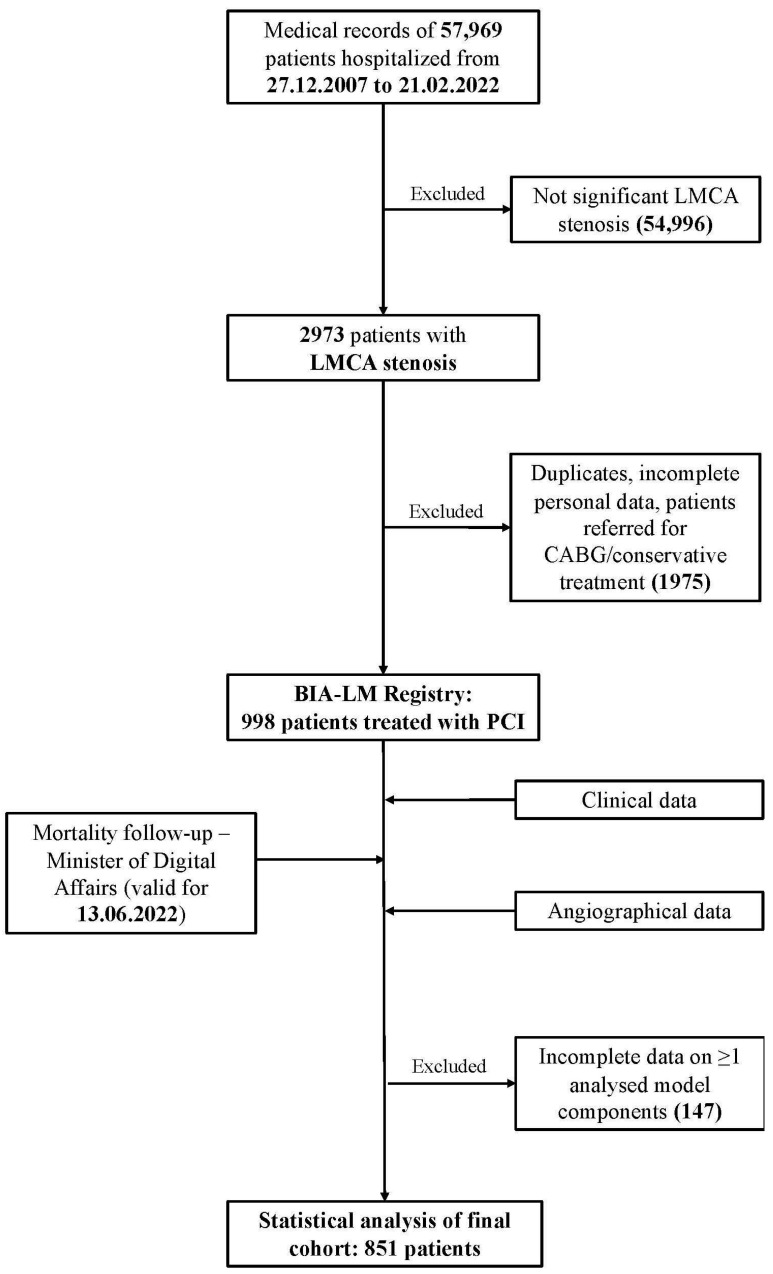
Flow chart of study design. CABG, coronary artery bypass grafting; LMCA, left main coronary artery disease; PCI, percutaneous coronary intervention.

**Figure 2 jcm-13-06907-f002:**
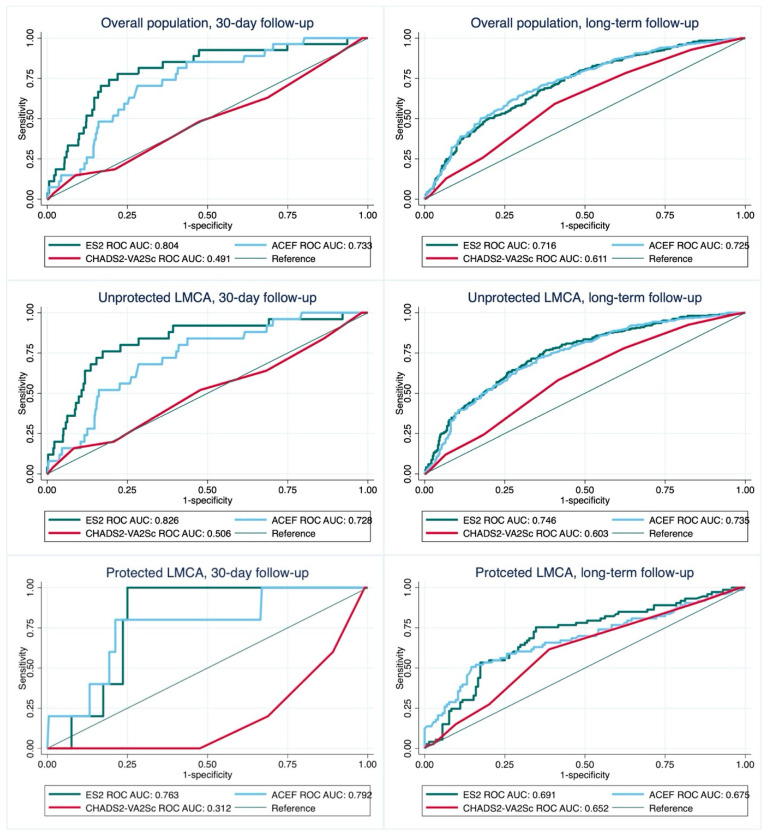
The receiver operating characteristic curve of EuroSCORE II, CHA_2_DS_2_-VASc, and ACEF score as a predictor of death. Abbreviations: AUC, area under curve; LMCA, left main coronary artery; ROC, receiver operating characteristic.

**Figure 3 jcm-13-06907-f003:**
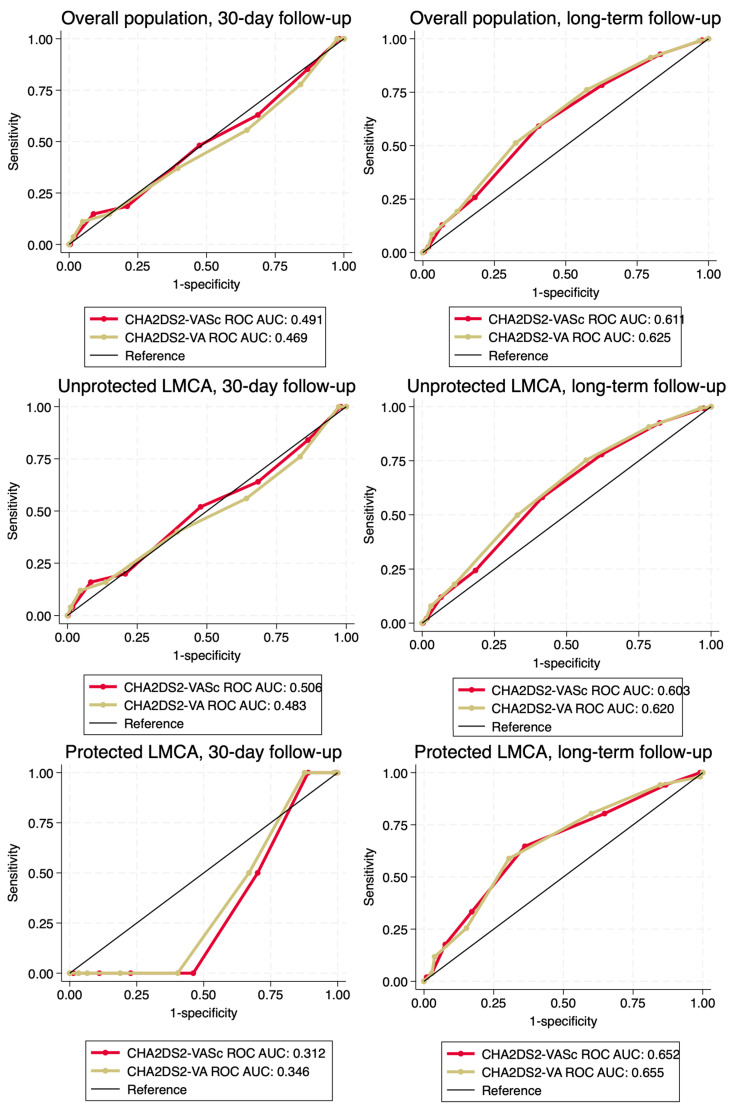
Comparison of CHA_2_DS_2_-VASc with CHA_2_DS_2_-VA receiver operating characteristic curves. Abbreviations: AUC, area under curve; LMCA, left main coronary artery; ROC, receiver operating characteristic.

**Figure 4 jcm-13-06907-f004:**
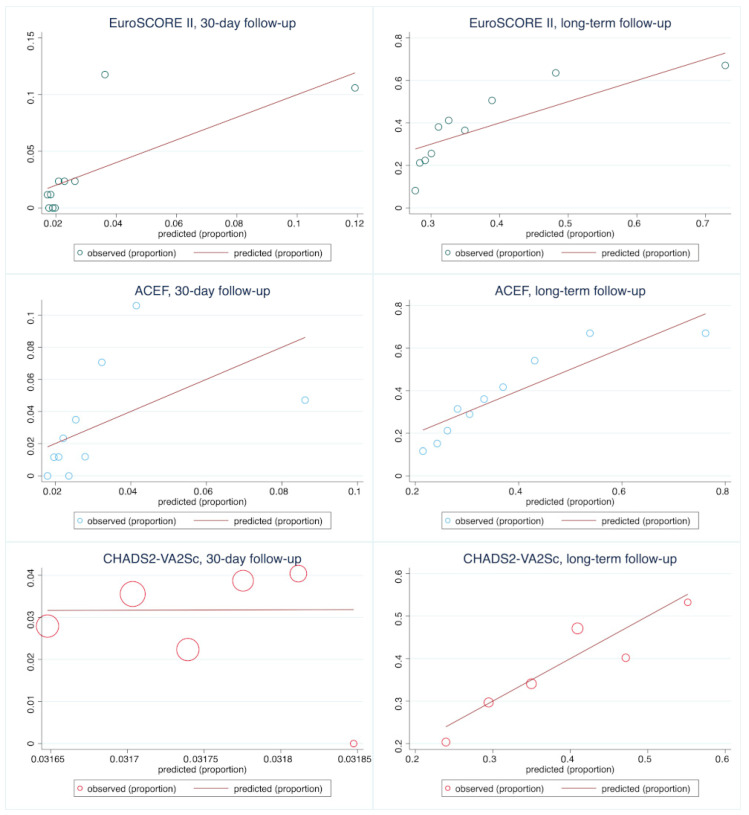
Calibration plots of the analyzed models in the overall population undergoing left main coronary artery PCI in short- and long-term observation.

**Table 1 jcm-13-06907-t001:** Baseline characteristics.

Variable	Overall Population	Unprotected LMCA	Protected LMCA	*p* Value
**Number of patients**	851 (100%)	695 (82%)	156 (18%)	N/A
**Baseline data**				
**Male**	625 (73.4%)	501 (72.1%)	124 (79.5%)	0.059
**Age (years)**	71 (63–79)	71 (63–79)	68 (63.5–77)	0.08
**Medical history**				
**Diabetes**	297 (35.4%)	231 (33.8%)	66 (42.3%)	0.04
**IDDM**	63 (7.4%)	51 (8%)	12 (7.7%)	0.90
**Hypertension**	721 (86.5%)	582 (85.6%)	139 (90.3%)	0.13
**Hypercholesterolemia**	763 (91.5%)	616 (90.6%)	147 (95.5%)	0.051
**AF**	201 (24.3%)	169 (25%)	32 (20.8%)	0.27
**KDIGO stage** **≥** **G3a**	271 (33%)	227 (34%)	44 (28.8%)	0.22
**Dialysis**	10 (1.2%)	10 (1.5%)	0 (0%)	0.13
**PAD**	226 (27.2%)	178 (26.3%)	48 (31.2%)	<0.22
**COPD**	68 (8.2%)	58 (8.6%)	10 (6.5%)	0.39
**EF (%)**	48 (36–55)	48 (36–55)	48 (35–55)	0.41
**Heart failure**	499 (59%)	399 (57.8%)	100 (64.1%)	<0.15
**Severe VHD**	42 (4.9%)	34 (4.9%)	8 (5.1%)	0.9
**Previous MI**	378 (46.8%)	283 (43%)	95 (63.3%)	<0.001
**Previous stroke**	81 (10.5%)	62 (10%)	19 (12.4%)	0.4
**Previous PCI**	358 (43.7%)	290 (43.2%)	68 (46%)	0.55
**Pulmonary hypertension**				
**Moderate (30–54 mmHg)**	74 (8.7%)	59 (9.3%)	15 (9.6%)	0.69
**Severe (≥55 mmHg)**	15 (1.8%)	13 (2%)	2 (1.3%)	0.53
**Biochemical tests**				
**Hemoglobin (g/dL)**	13.4 (12.2–14.4)	13.3 (12.2–14.3)	13.6 (12.3–14.5)	0.25
**Platelets (10^3^/mL)**	207 (172–251)	209 (174–253)	194 (158–245)	0.02
**Creatinine (mg/dL)**	0.97 (0.85–1.22)	0.98 (0.84–1.23)	0.97 (0.86–1.18)	0.83
**GFR (mL/min/1.73 m^2^)**	72.0 (54.0–87.1)	70.8 (53.4–86.9)	74.5 (56.8–88.6)	0.23
**Clinical presentation**				
**Chronic coronary syndrome**	493 (58.7%)	372 (54.3%)	121 (78.1%)	<0.001
**ACS**	347 (41.3%)	313 (45.7%)	34 (22%)	
**STEMI**	31 (4%)	30 (4.9%)	1 (0.7%)	0.003
**NSTEMI**	209 (27.5%)	192 (31.1%)	17 (11.9%)	
**UA**	89 (11.9%)	74 (11.1%)	15 (9.7%)	
**Clinical prediction models**				
**EuroSCORE II**	3.1 (5.4)	2.6 (4.2)	5.3 (7.7)	<0.001
**ACEF**	1.56 (0.9)	1.56 (0.9)	1.55 (0.75)	0.99
**CHA_2_DS_2_-VASc**	4 (2)	4 (2)	4 (2)	0.73
**CHA_2_DS_2_-VA**	4 (2)	4 (2)	4 (2)	0.73

Abbreviations: ACEF, age, creatinine, ejection fraction; ACS, acute coronary syndrome; AF, atrial fibrillation; COPD, chronic obstructive pulmonary disease; EF, ejection fraction; GFR, glomerular filtration rate; IDDM, insulin-dependent diabetes mellitus; LMCA, left main coronary artery; MI, myocardial infarction; NSTEMI, non-ST-elevation myocardial infarction; PAD, peripheral artery disease; PCI, percutaneous coronary intervention; STEMI, ST-elevation myocardial infarction; UA, unstable angina; VHD, valvular heart disease.

**Table 2 jcm-13-06907-t002:** Clinical prediction models’ discrimination in short- and long-term studied population.

CPM	AUC	Standard Error	95% CI	*p*-Value
**Overall population**				
**30 days**				
**EuroSCORE II**	0.804	0.044	0.717–0.890	<0.001
**ACEF**	0.733	0.043	0.648–0.817	
**CHA_2_DS_2_-VASc**	0.491	0.059	0.375–0.605	
**CHA_2_DS_2_-VA**	0.469	0.06	0.351–0.588	
**Long-term**				
**EuroSCORE II**	0.716	0.018	0.680–0.750	<0.001
**ACEF**	0.725	0.018	0.690–0.760	
**CHA_2_DS_2_-VASc**	0.611	0.019	0.573–0.648	
**CHA_2_DS_2_-VA**	0.625	0.019	0.588–0.662	
**Unprotected LMCA**				
**30 days**				
**EuroSCORE II**	0.826	0.045	0.737–0.913	<0.001
**ACEF**	0.728	0.047	0.636–0.818	
**CHA_2_DS_2_-VASc**	0.506	0.062	0.384–0.628	
**CHA_2_DS_2_-VA**	0.483	0.065	0.356–0.609	
**Long-term**				
**EuroSCORE II**	0.746	0.019	0.709–0.783	<0.001
**ACEF**	0.735	0.019	0.697–0.772	
**CHA_2_DS_2_-VASc**	0.603	0.021	0.561–0.644	
**CHA_2_DS_2_-VA**	0.620	0.021	0.579–0.661	
**Protected LMCA**				
**30 days**				
**EuroSCORE II**	0.763	0.058	0.648–0.877	<0.001
**ACEF**	0.792	0.033	0.727–0.857	
**CHA_2_DS_2_-VASc**	0.312	0.111	0.009–0.529	
**CHA_2_DS_2_-VA**	0.346	0.122	0.106–0.585	
**Long-term**				
**EuroSCORE II**	0.691	0.045	0.602–0.778	0.88
**ACEF**	0.675	0.049	0.579–0.769	
**CHA_2_DS_2_-VASc**	0.652	0.046	0.560–0.742	
**CHA_2_DS_2_-VA**	0.655	0.046	0.566–0.745	

Abbreviations: AUC, area under curve; CI, confidence interval; CPM, clinical prediction model; LMCA, left main coronary artery.

**Table 3 jcm-13-06907-t003:** Calibration of clinical prediction models in overall population.

Group	Upper Boundaries of Predicted Probabilities	Observed Deaths	Expected Deaths	Observed Alive	Expected Alive	Total Observations	*p*-Value
**EuroSCORE II**							
**30 days**							
**1**	0.018	1	1.5	85	84.5	86	0.006
**2**	0.018	0	1.5	85	83.5	85	
**3**	0.019	1	1.6	84	83.4	85	
**4**	0.019	0	1.6	86	84.4	86	
**5**	0.020	0	1.7	84	82.3	84	
**6**	0.022	2	1.8	83	83.2	85	
**7**	0.024	2	1.9	83	83.1	85	
**8**	0.029	2	2.2	83	82.8	85	
**9**	0.046	10	3.1	75	81.9	85	
**10**	0.552	9	10.1	76	74.9	85	
**Long-term**							
**1**	0.280	7	23.8	79	62.2	86	<0.001
**2**	0.288	18	24.1	67	60.9	85	
**3**	0.296	19	24.8	66	60.2	85	
**4**	0.305	22	25.8	64	60.2	86	
**5**	0.318	32	26.1	52	57.9	84	
**6**	0.336	35	27.7	50	57.3	85	
**7**	0.364	31	29.7	54	55.3	85	
**8**	0.419	43	33.1	42	51.9	85	
**9**	0.556	54	40.9	31	44.1	85	
**10**	0.983	57	61.9	28	23.1	85	
**ACEF**							
**30 days**							
**1**	0.019	0	1.5	86	84.5	86	0.01
**2**	0.020	1	1.7	85	84.3	86	
**3**	0.022	1	1.8	84	83.2	85	
**4**	0.023	2	1.9	84	84.1	86	
**5**	0.024	0	2	83	81	83	
**6**	0.026	3	2.2	83	83.8	86	
**7**	0.030	1	2.3	83	81.7	84	
**8**	0.035	6	2.7	79	82.3	85	
**9**	0.049	9	3.5	76	81.5	85	
**10**	0.517	4	7.3	81	77.7	85	
**Long-term**							
**1**	0.231	10	18.5	76	67.5	86	0.001
**2**	0.252	13	20.8	73	65.2	86	
**3**	0.272	18	22.3	67	62.7	85	
**4**	0.292	27	24.2	59	61.8	86	
**5**	0.317	24	25.3	59	57.7	83	
**6**	0.348	31	28.6	55	57.4	86	
**7**	0.398	35	31.1	49	52.9	84	
**8**	0.464	46	36.7	39	48.3	85	
**9**	0.615	57	45.7	28	39.3	85	
**10**	0.996	57	64.8	28	20.2	85	
**CHA_2_DS_2_-VASc**							
**30 days**							
**2**	0.032	5	5.7	174	173.3	179	0.80
**4**	0.032	8	7.1	217	217.9	225	
**6**	0.032	4	5.7	175	173.3	179	
**8**	0.032	6	4.9	149	150.1	155	
**9**	0.032	4	3.1	95	95.9	99	
**10**	0.032	0	0.4	14	13.6	14	
**Long-term**							
**1**	0.245	23	27.1	90	85.9	113	0.16
**3**	0.295	46	45.7	109	109.3	155	
**5**	0.350	61	62.6	118	116.4	179	
**7**	0.409	106	92.1	119	132.9	225	
**9**	0.472	41	48.1	61	53.9	102	
**10**	0.656	41	42.5	36	34.5	77	
**CHA_2_DS_2_-VA**							
**30 days**							
**1**	0.030	4	3.7	121	121.3	125	0.66
**3**	0.031	6	6.5	205	204.5	211	
**6**	0.032	5	6.7	207	205.3	212	
**8**	0.033	6	5.5	161	161.5	167	
**9**	0.034	6	3.9	110	112.1	116	
**10**	0.035	0	0.7	20	19.3	20	
**Long-term**							
**1**	0.236	28	31.1	108	104.9	136	0.27
**3**	0.296	48	49.4	119	117.6	167	
**6**	0.363	79	77.0	133	135.0	212	
**8**	0.437	102	92.1	109	118.9	211	
**9**	0.513	34	41.6	47	39.4	81	
**10**	0.661	27	26.9	17	17.1	44	

## Data Availability

The datasets used and analyzed during the current study are available from the corresponding author on reasonable request.

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
