# Peer review of "Validation of EuroSCORE II, ACEF Score, CHA2DS2-VASc, and CHA2DS2-VA in Patients Undergoing Left Main Coronary Artery Angioplasty: Analysis from All-Comers BIA-LM Registry"

_jcm, 2024, doi:10.3390/jcm13226907_

Round 1
Reviewer 1 Report
Comments and Suggestions for Authors
The authors of the manuscript present results from an original study, including 851 patients from the BIA-LM Registry who had underwent coronary artery bypass grafting (CABG). The aimed of this study was to validate the prognostic potential of 3 scores - EuroSCORE II, ACEF and CHA2DS2-VASc to predict short-term and long-term mortality rates in this cohort of patients. According to authors' results EuroSCORE II showed good mortality prediction for the short-term period of observation, but the predicted risk should be interpreted with caution due to poor calibration in their study. ACEF and EuroSCORE II may be useful for long-term mortality prediction. The manuscript is topical and important from clinical point of view. Coronary artery disease (CAD) is among the leading cause of death in many countries worlwide. Due to progress in both pharmacological and invasive - surgical and non-surgical treatment the rate of deaths due to CAD has declined significantly for the last 3 decades. With the development of new percutaneous catheters-based techniques and minimally invasive surgical methods for coronary revascularization many cardiologists face the dilema in some of their patients which approach - percutaneous coronary revascularization or surgical revascularization would be the most optimal approach to their patients. The decision is influenced by many factors and individualized for every patient. Stratification scores assessing the potential benefits and the risk of unfavorable outcomes durng and after the procedure facilitate the clinician's choice.
The manuscript is written very well. The text and the tables are plain, and the figures are illustrative. I have just one recommendation to the authors - to replace the dot in the title between the words "Angioplasty" and Analysis" by a colon.
Author Response
The authors of the manuscript present results from an original study, including 851 patients from the BIA-LM Registry who had underwent coronary artery bypass grafting (CABG). The aimed of this study was to validate the prognostic potential of 3 scores - EuroSCORE II, ACEF and CHA2DS2-VASc to predict short-term and long-term mortality rates in this cohort of patients. According to authors' results EuroSCORE II showed good mortality prediction for the short-term period of observation, but the predicted risk should be interpreted with caution due to poor calibration in their study. ACEF and EuroSCORE II may be useful for long-term mortality prediction. The manuscript is topical and important from clinical point of view. Coronary artery disease (CAD) is among the leading cause of death in many countries worlwide. Due to progress in both pharmacological and invasive - surgical and non-surgical treatment the rate of deaths due to CAD has declined significantly for the last 3 decades. With the development of new percutaneous catheters-based techniques and minimally invasive surgical methods for coronary revascularization many cardiologists face the dilema in some of their patients which approach - percutaneous coronary revascularization or surgical revascularization would be the most optimal approach to their patients. The decision is influenced by many factors and individualized for every patient. Stratification scores assessing the potential benefits and the risk of unfavorable outcomes durng and after the procedure facilitate the clinician's choice.
The manuscript is written very well. The text and the tables are plain, and the figures are illustrative. I have just one recommendation to the authors - to replace the dot in the title between the words "Angioplasty" and Analysis" by a colon.
Reply: We thank for the time and effort invested in reviewing our manuscript. We have changed the title accordingly.
Changes: Please see modified title of the paper.
Reviewer 2 Report
Comments and Suggestions for Authors
The study presents actual data with a lot of clinical implications.
The present study analyses data collected retrospectively from the BIA-LM Registry.
In brief, the registry is a single-centre database of LMCA PCI performed in the Department of Invasive Cardiology, Medical University of Bialystok, Poland analyzing patients referred for surgical or conservative treatment during a period of 4 years, from 12.27.2008 to 02.21.2022.
A final cohort of 851 individuals undergoing LMCA angioplasty were included into analysis. All analysed scores were recalculated for every patient enrolled in the study: EuroSCORE II using the interactive calculator (available at https://www.euroscore.org), CHA2DS2-VASc as sum of 1 point for diagnosis of heart failure, hypertension, age between 65-74 years, diabetes mellitus, vascular disease, female gender and 2 points for age ≥75 years and prior stroke or transient ischemic attack, and ACEF as age divided by ejection fraction plus one if serum creatinine was ≥2 mg/dl.
One mention I have concerning CHA2DS2-VASC Score. Since 30 august 2024 is not actual. After 2024ESC Guideline for atrial fibrillation, CHA2DS2-VA Score is used instead of CHA2DS2-VASc. https://doi.org/10.1093/eurheartj/ehae176
CHA2DS2-VA = Congestive heart failure, hypertension, age ≥75 years (2 points), diabetes mellitus, prior stroke/transient ischaemic attack/arterial thromboembolism (2 points), vascular disease, age 65–74 years (score)
Analyses were performed for the overall population, patients with unprotected LMCA (i.e., with no previous CABG) and protected LMCA (patients with a history of previous CABG). Further calculations were performed based on the severity of CAD. Patients were divided into four subgroups: isolated LMCA narrowing, with additional 1-, 2- or 3-vessel disease (1-VD, 2-VD, 3-VD, respectively).
The design of the study is very well conceived.
In the overall cohort, EuroSCORE II as compared to ACEF and CHA2DS2-VASc, provided the best discriminative performance for the 30-day mortality (P=0.03 and P<0.001). In the long-term, EuroSCORE II and ACEF both were good mortality predictors (P=0.53 145 for comparison) with better performance than CHA2DS2-VASc (P<0.001 for both two model comparisons).
In patients with unprotected LMCA, EuroSCORE II showed better discriminative performance than ACEF and CHA2DS2-VASc in short-term (P=0.051 and P<0.001), while in the long-term EuroSCORE II and ACEF were better than CHA2DS2-VASc (P<0.001 for both two model comparisons). In the subgroup of post-CABG patients, EuroSCORE II and ACEF provided moderate 30-day mortality discriminative performance, as opposed to poor CHA2DS2-VASc discrimination. In the long-term, all of the analysed CPMs provided acceptable discriminative performance.
Pertinent conclusions.
Author Response
The study presents actual data with a lot of clinical implications.
The present study analyses data collected retrospectively from the BIA-LM Registry.
In brief, the registry is a single-centre database of LMCA PCI performed in the Department of Invasive Cardiology, Medical University of Bialystok, Poland analyzing patients referred for surgical or conservative treatment during a period of 4 years, from 12.27.2008 to 02.21.2022.
A final cohort of 851 individuals undergoing LMCA angioplasty were included into analysis. All analysed scores were recalculated for every patient enrolled in the study: EuroSCORE II using the interactive calculator (available at https://www.euroscore.org), CHA2DS2-VASc as sum of 1 point for diagnosis of heart failure, hypertension, age between 65-74 years, diabetes mellitus, vascular disease, female gender and 2 points for age ≥75 years and prior stroke or transient ischemic attack, and ACEF as age divided by ejection fraction plus one if serum creatinine was ≥2 mg/dl.
One mention I have concerning CHA2DS2-VASC Score. Since 30 august 2024 is not actual. After 2024ESC Guideline for atrial fibrillation, CHA2DS2-VA Score is used instead of CHA2DS2-VASc. https://doi.org/10.1093/eurheartj/ehae176
CHA2DS2-VA = Congestive heart failure, hypertension, age ≥75 years (2 points), diabetes mellitus, prior stroke/transient ischaemic attack/arterial thromboembolism (2 points), vascular disease, age 65–74 years (score)
Analyses were performed for the overall population, patients with unprotected LMCA (i.e., with no previous CABG) and protected LMCA (patients with a history of previous CABG). Further calculations were performed based on the severity of CAD. Patients were divided into four subgroups: isolated LMCA narrowing, with additional 1-, 2- or 3-vessel disease (1-VD, 2-VD, 3-VD, respectively).
The design of the study is very well conceived.
In the overall cohort, EuroSCORE II as compared to ACEF and CHA2DS2-VASc, provided the best discriminative performance for the 30-day mortality (P=0.03 and P<0.001). In the long-term, EuroSCORE II and ACEF both were good mortality predictors (P=0.53 145 for comparison) with better performance than CHA2DS2-VASc (P<0.001 for both two model comparisons).
In patients with unprotected LMCA, EuroSCORE II showed better discriminative performance than ACEF and CHA2DS2-VASc in short-term (P=0.051 and P<0.001), while in the long-term EuroSCORE II and ACEF were better than CHA2DS2-VASc (P<0.001 for both two model comparisons). In the subgroup of post-CABG patients, EuroSCORE II and ACEF provided moderate 30-day mortality discriminative performance, as opposed to poor CHA2DS2-VASc discrimination. In the long-term, all of the analysed CPMs provided acceptable discriminative performance.
Pertinent conclusions.
Reply: We thank for the kind words and remark regarding recent guidelines recommendations to use CHA2DS2-VA instead of CHA2DS2-VASc. To make our analysis up to date, we have now included CHADS-VA analyses in the manuscript. However, to remain maximal generalizability, we decided not to remove results investigating validation of CHA2DS2-VASc score – calculator widely used for the past 14 years.
Now, readers can find information on the performance of four calculators in present manuscript.
Changes: Please see changes in the Introduction, Methods, Results and Discussion section and new Figure 3 now included in the manuscript.
Reviewer 3 Report
Comments and Suggestions for Authors
In the study the authors established that EuroSCORE II showed good mortality prediction the short-term observation, its predicted risk should be interpreted with caution due to poor calibration. ACEF score and EuroSCORE II may be useful in long-term mortality prediction.
Although the findings are impressive and practically useful, I would like to make some comments.
1. The authors might discuss the patients-related factors incorporated into the EuroSCORE II, which had not been added to Table 1.
2. ACEF score was developed for 30-day mortality after elective or emergency cardiac surgery, so it is not suitable for the long-term risk evaluation. Please, explaine this.
3. CHAâ‚‚DSâ‚‚-VASc Score is avaiable for Atrial Fibrillation Stroke Risk, so that is not corresponded to the aim of the study.
4. The authors should explaine how many patients with AF were included in the analysis and why they used CHAâ‚‚DSâ‚‚-VASc Score for the evaluation of clinical outcomes after PCI.
5. Please, extend the section Discussion and Conclusio taking into consideration the issues mentioned above.
Author Response
In the study the authors established that EuroSCORE II showed good mortality prediction the short-term observation, its predicted risk should be interpreted with caution due to poor calibration. ACEF score and EuroSCORE II may be useful in long-term mortality prediction.
Although the findings are impressive and practically useful, I would like to make some comments.
- The authors might discuss the patients-related factors incorporated into the EuroSCORE II, which had not been added to Table 1.
Reply 1: Thank you for this important suggestion.
In our dataset we gathered all of the variables initially included in EuroSCORE II calculator. Actually, most of the variables incorporated into the EuroSCORE II are already included in Table 1., although some of them are not directly titled as in the original publication by Nashef et al. [1], e.g., patients with severe valvular disease were considered either 2- or more procedures, patients with STEMI were considered emergency procedure. Due to the low prevalence of selected variables, some of them were not included in the Table 1., however, now as the Reviewer has suggested, we have updated Table 1. with new selected data included in EuroSCORE II calculator.
[1] Samer A.M. Nashef, François Roques, Linda D. Sharples, Johan Nilsson, Christopher Smith, Antony R. Goldstone, Ulf Lockowandt, EuroSCORE II, European Journal of Cardio-Thoracic Surgery, Volume 41, Issue 4, April 2012, Pages 734–745, https://doi.org/10.1093/ejcts/ezs043
Changes 1: Please see Table 1. updated.
- ACEF score was developed for 30-day mortality after elective or emergency cardiac surgery, so it is not suitable for the long-term risk evaluation. Please, explaine this.
Reply 2: Thank you for the comment. Indeed, ACEF Score developed by Ranucci was initially used for in-hospital and 30-day after elective cardiac surgery moratlity prediction. It’s performance in this setting was validated multiple times before. Thus, the aim of our study was to seek for novel application of the well-known clinical prediction models.
We investigated both short- and long-term prognostic value in patients undergoing PCI for significant LMCA stenosis. In the short-term, ACEF provided worse mortality prediction than EuroSCORE II, however, in the long-term observation ACEF showed the best discriminative performance amongst analysed calculators. Good long-term performance of ACEF score suggests that universal three factors may act as a surrogate for comorbidity burden and general health status. This may have clinical implications, as patients with higher ACEF scores may obtain more personalized post-procedure surveillance. Higher-risk patients can be identified for closer monitoring, targeted follow-up, and intensive secondary and tertiary prevention, such as intensification of guideline-directed medical therapy targeted at nephroprotection and heart failure with mildly reduced or reduced ejection fraction – treatment including ACE inhibitors, SLGT-2 inhibitors, and finerenone in selected individuals.
Changes 2: Please see Discussion section modified.
- CHAâ‚‚DSâ‚‚-VASc Score is avaiable for Atrial Fibrillation Stroke Risk, so that is not corresponded to the aim of the study.
Reply 3: Thank you for this comment, we are glad to clarify this issue. In fact, none of the three analysed calculators were initially developed for assessment of adverse cardiovascular events in patients undergoing coronary angioplasty. A number of CPMs were developed in order to facilitate decision-making in patients with coronary artery disease undergoing PCI, including anatomic SYNTAX, Mayo Clinic and NCDR or, combining anatomic conditions with clinical data SYNTAX II. However, the inclusion of many variables leading to higher accuracy at the expense of complexity and less bedside usefulness limits the clinical utility.
Thus, we included three, arguably most commonly used cardiovascular clinical prediction models. We found that EuroSCORE II and ACEF may be especially useful in short- and long-term mortality prediction. When it comes to CHA2DS2-VASc, its discriminative performance was inferior to the aforementioned calculators. However, it showed good calibration in most of the subgroups suggesting that it’s thromboembolic risk score, actually indicating co-morbidity burden severity, may be useful when assessing low- or high adverse event risk patients, especially on the populational level.
Changes 3: Please see Discussion section modified.
- The authors should explaine how many patients with AF were included in the analysis and why they used CHAâ‚‚DSâ‚‚-VASc Score for the evaluation of clinical outcomes after PCI.
Reply 4: Thank you for this comment. The data for rates of concomitant atrial fibrillation are provided in Table 1. As we elaborated above, we included three widely adopted risk scores and sought to analyse their performance in short- and long-term mortality prediction amongst patients undergoing left main coronary artery PCI.
The models were validated in various clinical settings outside of their initial purpose, however, whether these risk scores may have short- and long-term prognostic value in patients undergoing PCI for significant LMCA stenosis remains to be unknown. Thus, aiming the gap in evidence, the purpose of study was to assess relation between EuroSCORE II, CHA2DS2-VASc, ACEF and mortality in the overall population of patients undergoing LMCA angioplasty. Moreover, our analyses included stratification on the severity and extent of CAD, and based on the history of previous revascularization.
Changes 4: Please see Discussion section modified.
- Please, extend the section Discussion and Conclusio taking into consideration the issues mentioned above.
Reply 5: Thank you for the valuable comments. As suggested, we have addressed them in the discussion section.
Changes 5: Please see Discussion section modified accordingly.
Round 2
Reviewer 3 Report
Comments and Suggestions for Authors
The authors resubmitted the article after critical revision. The respond to the reviewers are clear and readable. The article in its revised version has logical structure, contains legible figures and tables and thoroughly prepated discussion. I have no serious conserns about the paper in its revised version.